# Application of geographically weighted regression analysis to assess predictors of short birth interval hot spots in Ethiopia

Desalegn Markos Shifti [1,2]*, Catherine Chojenta[2], Elizabeth G. Holliday[3], Deborah Loxton[2]

**1** Saint Paul's Hospital Millennium Medical College, Addis Ababa, Ethiopia, **2** School of Medicine and Public Health, Priority Research Centre for Generational Health and Ageing, University of Newcastle, Newcastle, New South Wales, Australia, **3** School of Medicine and Public Health, Centre for Clinical Epidemiology and Biostatistics, University of Newcastle, Newcastle, New South Wales, Australia

* E-mail: desalegnmarkos@gmail.com, desalegnmarkos.shifti@uon.edu.au

## Abstract

### Background

Birth interval duration is an important and modifiable risk factor for adverse child and maternal health outcomes. Understanding the spatial distribution of short birth interval, an interbirth interval of less than 33 months, and its predictors are vital to prioritize and facilitate targeted interventions. However, the spatial variation of short birth interval and its underlying factors have not been investigated in Ethiopia.

### Objective

This study aimed to assess the predictors of short birth interval hot spots in Ethiopia.

### Methods

The study used data from the 2016 Ethiopia Demographic and Health Survey and included 8,448 women in the analysis. The spatial variation of short birth interval was first examined using hot spot analysis (Local Getis-Ord Gi* statistic). Ordinary least squares regression was used to identify factors explaining the geographic variation of short birth interval. Geographically weighted regression was used to explore the spatial variability of relationships between short birth interval and selected predictors.

### Results

Statistically significant hot spots of short birth interval were found in Somali Region, Oromia Region, Southern Nations, Nationalities, and Peoples' Region and some parts of Afar Region. Women with no education or with primary education, having a husband with higher education (above secondary education), and coming from a household with a poorer wealth quintile or middle wealth quintile were predictors of the spatial variation of short birth interval. The predictive strength of these factors varied across the study area. The geographically

**Data Availability Statement:** Data underlying the study was sourced from a third party. The data used in this study is available in The DHS Program database, which can be accessed with registration

in https://dhsprogram.com/data/new-user-registration.cfm (i.e., New User Registration platform). The authors confirm that they did not have any special access to this data which others would not have.

**Funding:** The authors received no specific funding for this work.

**Competing interests:** The authors have declared that no competing interests exist.

weighted regression model explained about 64% of the variation in short birth interval occurrence.

## Conclusion

Residing in a geographic area where a high proportion of women had either no education or only primary education, had a husband with higher education, or were from a household in the poorer or middle wealth quintile increased the risk of experiencing short birth interval. Our detailed maps of short birth interval hot spots and its predictors will assist decision makers in implementing precision public health.

## Introduction

A birth interval is defined as the length of time between two successive live births to the same mother [1, 2]. The World Health Organization (WHO) recommends a birth-to-pregnancy interval of at least 24 months, which is equivalent to a 33-month birth-to-birth interval, to reduce the risk of adverse maternal and child health outcomes [3].

A short birth interval is associated with a higher risk of maternal morbidities, including miscarriage, preeclampsia, high blood pressure, and premature rupture of membranes [4, 5]. It is also associated with an increased risk of small size for gestational age [6], low birth weight [6–8], preterm birth [6, 7], congenital anomalies [9, 10], autism [11] and infant mortality [6, 12].

Birth interval practices vary widely around the world [13]. A study conducted in the Democratic Republic of Congo (DRC) identified statistically significant spatial variation in short birth interval among women across provinces [14]. The prevalence of short birth interval among regions of Ethiopia ranges from 10.6% in Amhara Region to 45.9% in Somali Region [15]. Moreover, despite the increasing trend in contraceptive utilization in Ethiopia, from 5.9% in 2000 to 14.0% in 2005, 27.0% in 2011 and 35.0% in 2016, the prevalence of short birth interval has remained unchanged, 19.7% in 2000, 21.4% in 2005, 20.5% in 2011 and 21.7% in 2016 [15–18].

To the best of our knowledge, no previous study has assessed spatial variation in short birth interval, or predictors of such spatial variation among women in Ethiopia. This implies that the 'where' aspect of short birth interval has been inadvertently missed, leading to a significant evidence lacuna. Failure to consider geographic variation in short birth interval, whether it is random, dispersed, or clustered, may result in socioeconomic inequity in public health resource allocation and policy design. Since birth intervals are amenable to policy intervention [19], it is valuable to identify high-risk geographic areas for targeted intervention development.

Empirical evidence has documented several risk factors of short birth intervals. These include young maternal age [20–22], maternal education [2, 21, 23–25], household wealth status [23, 26], rural place of residence [24, 26], having fewer than 4 children [25, 27], preceding child being female [22, 23, 25, 27], death of preceding child [27, 28], duration of breastfeeding of less than 24 months [23, 25], and non-use of modern contraceptives [23–27]. Nevertheless, predictors of spatial variation of short birth interval have not been well understood in Ethiopia.

National policies regarding birth control that aim to increase contraceptive uptake (from 42% to 55%) and reduce the total fertility rate (from 4.1 to 3) by the end of 2020 [29], both of

which affect birth interval duration, may not impact all regions equally. Regional-level estimates in short birth interval are needed to inform policy makers about populations where the burden is high. A precision public health approach can help identify vulnerable pockets of the population that most require increased efforts to prevent short birth interval [30, 31].

The current study aimed to assess predictors of short birth interval hot spots in Ethiopia. The findings of this study will help policy makers, program planners, and other stakeholders in guiding health investment, and prioritize prevention and intervention programs. In addition, mapping short birth interval hot spots will provide a deeper understanding of the impact of past intervention strategies, including family planning services in each region of the country.

## Methods

### Study setting and design

The study was conducted in Ethiopia, which is located in the Horn of Africa ($3^0$–$15^0$ N latitude and $33^0$–$48^0$ E longitude) [15]. The country occupies an area of 1.1 million square kilometres with an altitude that ranges from the highest peak at Ras Dashen (4,620 metres above sea level) down to the Dallol Depression, about 148 metres below sea level [32, 33]. Administratively, Ethiopia is divided into nine regions and two administrative cities [15].

This analysis was based on the 2016 Ethiopia Demographic and Health Survey (EDHS) data. The EDHS sample was derived using a stratified, two-stage cluster design where Enumeration Areas (EAs) were the sampling units for the first stage and households for the second stage. The detailed methodologies of the surveys are presented in the full EDHS report [15]. The current study included 8,448 women from 620 clusters, who had reported at least two live births during the five years preceding the 2016 survey. Women who had never been married (n = 12) were not included in the study since women who have multiple births out of wedlock are unlikely to plan their births in the same way as married women. When women had more than two births in the five years preceding the survey, birth interval of their most recent two births (i.e., the birth interval between the index child and the immediately preceding child) was uniformly considered for all the study participants.

Global Positioning System (GPS) receivers were used to collect the location data (geographic coordinates) of each survey cluster. The GPS reading was made at the centre of each cluster. The GPS data collectors ensured the centre was relatively open, away from tall buildings, and out from under tree canopy in order to receive adequate satellite signal strength. To maintain respondents' confidentiality, GPS latitude/longitude positions for all survey clusters were randomly displaced. The maximum displacement for urban clusters was two kilometres (km) and five km for 99% of rural clusters. The remaining 1% of the rural clusters were displaced a maximum of 10 km. The displacement was restricted to the country's second administrative level (DHS survey region) so that the points stay within the country [34]. In addition, the administrative polygons of Ethiopia, which were obtained from the Natural Earth [35] has been used to develop the map of hot and/or cold spots of short birth interval. The country's administrative polygons reflect administrative boundaries, such as regions, zones, and districts of Ethiopia.

### Study variables

The outcome variable, short birth interval, was defined as an interval of less than 33 months between two successive live births [3]. Women's birth interval data were collected through reviewing the date of birth of their biological children from children's birth /immunization certificate and/or asking information regarding their children's date of birth from the women.

Birth interval data of women for all their children born live irrespective of their survival status at the time of the interview were collected. For children who had birth certificates, their mothers were asked to confirm the accuracy of the information prior to documenting children's date of birth. This was done to avoid errors because in some cases the information on the document may be the date when the birth was recorded and not the date when the child was born. When children did not have a birth certificate, information regarding their date of birth were obtained from their mothers. Then, the length of birth interval was computed in months and the data were accessible for further analysis in this form. Further explanation about how birth interval data were collected can be found in the Demographic and Health Survey Interviewer's Manual [36].

The candidate explanatory variables included in the Exploratory Regression of the current study are presented online (see S1 Table). These were maternal age at first marriage, maternal age at birth of the preceding child, polygyny status, maternal education level, husband's/partner's education level, maternal occupation, husband's/partner's occupation, wealth quintile, sex of the preceding child, survival status of the preceding child, total number of children born before the index child, exposure to mass media, and perceived distance to the health facility. Variables were selected based on reviewed literature [2, 14, 20–28]. An Exploratory Regression tool, discussed below under the spatial regression analysis section, was used to identify properly specified Ordinary Least Squares (OLS) models.

## Data management and analysis

Descriptive analyses were performed using Stata version 14 statistical software *(StataCorp. Stata Statistical Software*: *Release 14. College Station*, *TX*: *StataCorp LP. 2015)*. The spatial analysis was performed using ArcGIS 10.3.1*(ESRI. ArcGIS Desktop*: *Release 10. Redlands*, *CA*: *Environmental Systems Research Institute. 2011)*. Before performing spatial analysis, the weighted proportion (using sample weight) of short birth interval and candidate explanatory variables (see S1 Table) data were exported to ArcGIS. A detailed explanation of the weighting procedure can be found elsewhere [15].

## Descriptive analysis

Participant characteristics were described using frequency with percent. Pearson's chi-squared tests were used to assess differences in short birth interval frequencies between place (urban/rural) and regions of residence.

## Spatial analysis

The global Moran's I statistic was computed to test for the presence of spatial autocorrelation. This statistic indicates whether the pattern of short birth interval in the study area is clustered, dispersed, or random. When the z-score or p-value indicates statistical significance, a positive Moran's I index value indicates a tendency toward clustering while a negative Moran's I index value indicates a tendency toward dispersion. Based on this, a decision was made about whether to reject the null hypothesis that short birth intervals are randomly distributed across the study area [37].

The Getis-Ord General G statistic was used to measure the degree of clustering, which may be high or low. The higher (or lower) the z-score, the stronger the intensity of the clustering. A z-score near zero indicates no apparent clustering within the study area. A positive z-score indicates clustering of high values and a negative z-score indicates clustering of low values [38].

Subsequently, Incremental Spatial Autocorrelation was assessed to calculate an appropriate distance threshold for identifying spatial processes that promote clustering [39]. Hot spot analysis using local Getis-Ord Gi* statistics [40] was used to depict short birth interval variation in the study area. This statistic produces a hot and/or cold spot map using short birth interval rate as the input. It compares the local mean rate (the rates for a cluster and its nearest neighboring clusters) to the global mean rate (the rates for all clusters). A z-score and p-value are produced for each cluster, allowing assessment of the significance of differences between local and global means. A high positive z-score and a small p-value for a feature (cluster in this case) indicate a spatial clustering of high values (a hot spot). A low negative z-score and a small p-value indicate a spatial clustering of low values (a cold spot). A z-score near zero indicates no apparent spatial clustering [40–43]. Getis-Ord Gi* statistic is given as [42]:

$$G_i^* = \frac{\sum_{j=1}^n w_{i,j} x_j \quad - \quad \hat{X}}{S \sqrt{\frac{\left[ n \sum_{j=1}^n w_{i,j}^2 \quad - \quad \left(\sum_{j=1}^n w_{i,j}\right)^2 \right]}{n-1}}}$$

where $x_j$ is the attribute value for feature (cluster in the current study) $j$, $w_{i,j}$ is the spatial weight between feature $i$ and $j$, $n$ is equal to the total number of features and

$$\hat{X} \quad = \quad \frac{\sum_{j=1}^n x_j}{n}$$

$$S \quad = \quad \sqrt{\frac{\sum_{j=1}^n x_j^2}{n} \quad - \quad (\hat{X})^2}$$

When estimating local Getis-Ord Gi* statistics, a False Discovery Rate (FDR) correction method was applied to account for multiple, dependent tests [44–46]. This helps to identify true clusters by estimating the number of false positives for a given confidence level and adjusting the critical p-value accordingly. Thus, statistically significant p-values are ranked from smallest (strongest) to largest (weakest), and based on the false positive estimate, the weakest are removed from the list [44, 46]. The importance of considering the FDR correction method in DHS data has been documented elsewhere [45]. The Ethiopian Polyconic Projected Coordinate System, based on the World Geodetic System 84 (WGS84) coordinate reference system (CRS), was used to produce a flattened map of the country.

## Spatial regression

**Ordinary least squares (OLS) regression.**    After identifying short birth interval hot spots, spatial regression modeling was performed to identify predictors of the observed spatial patterns of short birth interval. Findings from ordinary least squares (OLS) regression are only reliable if the regression model satisfies all of the assumptions that are required by this method [47]. The coefficients of explanatory variables in a properly specified OLS model should be statistically significant and have either a positive or negative sign. In addition, there should not be redundancy among explanatory variables (free from multicollinearity). The model should be unbiased (heteroscedasticity or non-stationarity). The residuals should be normally distributed and revealed no spatial patterns. The model should include key explanatory variables. The residuals must be free from spatial autocorrelation [47–49]. The OLS regression equation [50] is given as:

$$y_i \quad = \quad \beta_0 + \sum_{k=1}^p \beta_k x_{ik} + \varepsilon_i$$

where $i = 1,2,\ldots$n; $\beta_0, \beta_1, \beta_2, \ldots \beta_p$ are the model parameters, $y_i$ is the outcome variable for observation $i$, $x_{ik}$ are explanatory variables and $\varepsilon_1, \varepsilon_2, \ldots \varepsilon_n$ are the error term/residuals with zero mean and homogenous variance $\sigma^2$.

To identify a model that fulfills the assumption of the OLS method, Exploratory Regression, a data-mining tool, was used. Similar to Stepwise Regression, Exploratory Regression identifies models with high Adjusted $R^2$ values. Moreover, unlike Stepwise Regression, Exploratory Regression identifies models that meet all of the assumptions of the OLS method [47, 51, 52]. The model was validated using internal cross-validation. Cross-validation provides an idea of how well a model built in the training dataset predicts unknown values in a validation dataset. For a model that provides accurate predictions, the mean error should be close to 0, the root-mean-square error and average standard error should be as small as possible (this is useful when comparing models), and the root-mean-squared standardized error (RMSE) should be close to 1 [53]. The model in the current study fulfilled the above statistical requirements.

**Geographically weighted regression (GWR).** A variable that is a strong predictor in one cluster may not necessarily be a strong predictor in another cluster. This type of cluster variation (non-stationarity) can be identified through the use of GWR. In this context, GWR can help to answer the question: "Does the association vary across space?" Unlike OLS that fits a single linear regression equation to all of the data in the study area, GWR creates an equation for each DHS cluster. While the equation in OLS is calibrated using data from all features (cluster in this case), GWR uses data from nearby features. Thus, the GWR coefficient takes different values for each cluster [54, 55]. Maps of the coefficients associated with each explanatory variable, which are produced using the GWR, provide guidelines for targeted interventions. The GWR model [56] can be written as:

$$ y_i \quad = \quad \beta_0 \left( u_i v_i \right) + \sum\nolimits_{k=1}^{p} \beta_k (u_i, v_i) x_{ik} + \varepsilon_i $$

where $y_i$ are observations of response y, $(u_i v_i)$ are geographical points (longitude, latitude), $\beta_k(u_i v_i)$ $(k = 0, 1, \ldots p,)$ are $p$ unknown functions of geographic locations $(u_i v_i)$, $x_{ik}$ are explanatory variables at location $(u_i v_i)$, $i = 1,2,\ldots$n and $\varepsilon_i$ are error terms/residuals with zero mean and homogenous variance $\sigma^2$. Fig 1 presents a summary of the model's framework.

## Ethical statement

Ethical approval was obtained from the Human Research Ethics Committee (H-2018-0332), The University of Newcastle. The 2016 EDHS was approved by the National Research Ethics Review Committee of Ethiopia (NRERC) and ICF Macro International. Permission from The DHS Program was obtained to access the datasets.

## Results

Table 1 presents the weighted proportion of short birth interval based on place of residence and regions. Among women who experienced short birth interval, 94.0% resided in a rural part of the country. Similarly, the majority of women with short birth interval lived in Oromia Region (50.9%) and Southern Nations, Nationalities, and People's Region (SNNPR; 21.5%; Table 1). The overall prevalence of short birth interval in Ethiopia was 45.8% (95% CI: 42.91–48.62).

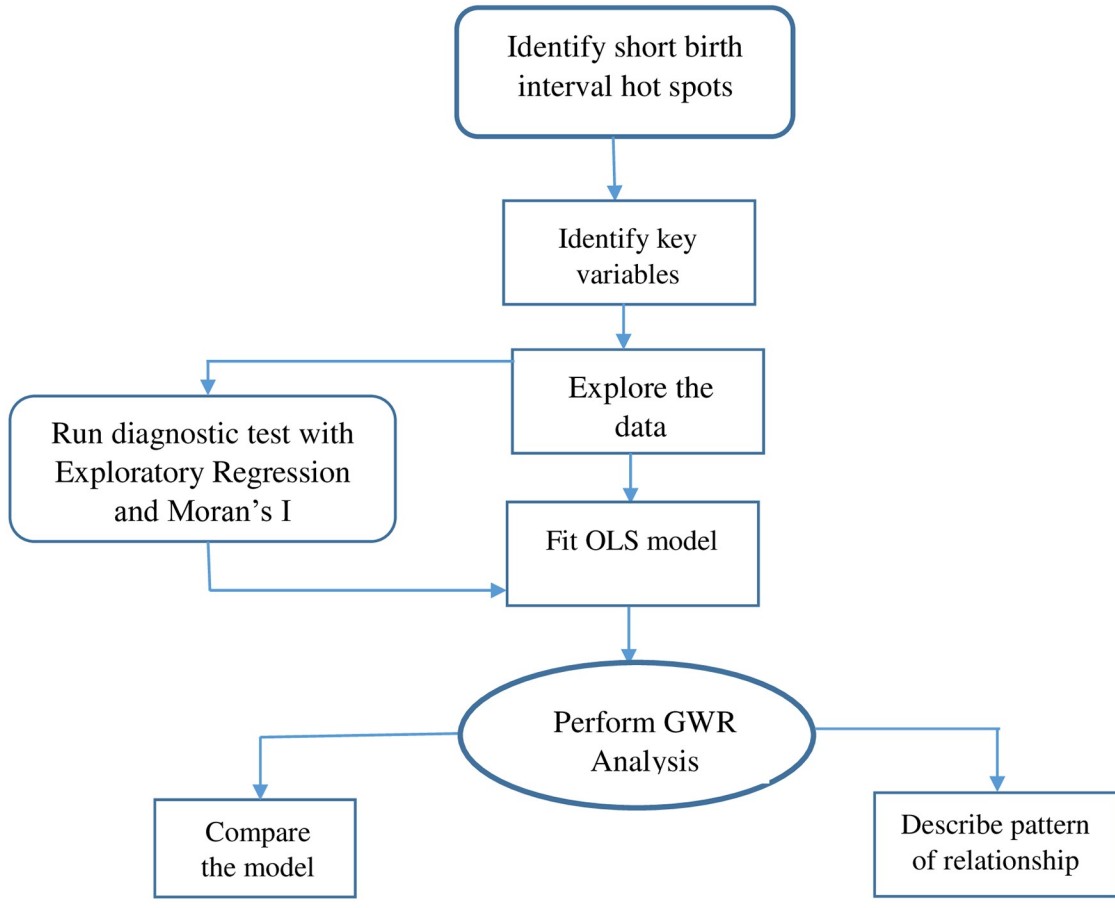

**Fig 1. Summary of models framework.** *OLS* = Ordinary Least Squares; *GWR* = Geographically Weighted Regression.

### Statistically significant hot spots

Fig 2 shows the spatial variation of short birth interval in Ethiopia. There was statistically significant (Global Moran's I = 0.260590, p-value <0.001) clustering of short birth interval in the study area. The Getis-Ord General G statistic revealed the presence of high clustering (z-score = 4.328, p-value<0.001). The average distance at which a feature (cluster in this case) has at least one neighbour was 18 kilometres (km). The maximum distance at which clustering of short birth interval rate peaked was at 122 km. Statistically significant hot spots of short birth interval were observed in eastern Ethiopia (Somali Region and a few parts of Afar Region), northern Ethiopia (some parts of Oromia Region), and northwestern Ethiopia (some parts of SNNPR). On the other hand, statistically significant cold spots of short birth interval were found in Tigray Region, Amhara Region, Addis Ababa, Benishangul-Gumuz, western Oromia and some parts of Afar Region (Fig 2).

### Factors affecting the spatial variation of short birth interval

Table 2 illustrates the results from an OLS model of short birth interval. This model explained about 57.0% of the variation in short birth interval (Adjusted $R^2$ = 0.57) and met all requirements of the OLS method [47, 51, 52]. While the Joint Wald Statistic indicated the overall model significance (p < 0.01), the robust probabilities showed coefficient significance (p < 0.01) for the explanatory variables. There was no problem of redundancy

**Table 1. Weighted proportion of short birth interval by place of residence and regions, EDHS 2016.**

| Variables | Weighted proportion | | P-value |
| --- | --- | --- | --- |
| | Non-short birth interval | Short birth interval | |
| | N (%) | N (%) | |
| **Place of residence** | | | |
| Urban | 818 (11.3) | 468 (6.0) | <0.001 |
| Rural | 3443 (88.7) | 3719 (94.0) | |
| **Regions** | | | |
| Tigray | 525 (7.5) | 263 (4.3) | <0.001 |
| Afar | 294 (0.7) | 534 (1.4) | |
| Amhara | 578 (25.2) | 222(11.1) | |
| Oromia | 641(39.6) | 666 (50.9) | |
| Somali | 343 (2.4) | 940 (8.0) | |
| Benishangul-Gumuz | 363 (1.0) | 372 (1.2) | |
| SNNPR* | 566 (20.7) | 478 (21.5) | |
| Gambella | 338 (0.3) | 219 (0.2) | |
| Harari | 231(0.2) | 225 (0.2) | |
| Addis Ababa | 188 (2.1) | 64 (0.8) | |
| Dire Dawa | 194 (0.3) | 204 (0.4) | |

*SNNPR = Southern Nations, Nationalities, and People's Region, EDHS = Ethiopia Demographic and Health Survey.

(multicollinearity) among explanatory variables as evidenced by the low variance inflation factor values (VIF, which was < 7.5). The Spatial Autocorrelation (Moran's I) test revealed that residuals were not spatially autocorrelated (p > 0.1). The Jarque-Bera statistic was non-significant (p > 0.10) indicating that the model residuals were normally distributed (Table 2). Predictors of short birth interval hot spots were women with no education or with primary

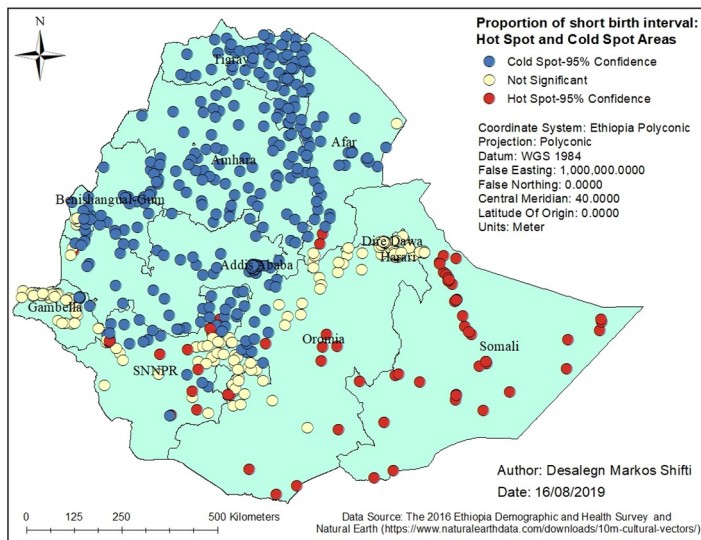

**Fig 2. Hot spots of short birth interval in Ethiopia, EDHS 2016.** *SNNPR* = Southern Nation, Nationalities and People's region; *EDHS* = Ethiopia Demographic and Health Survey.

**Table 2. Summary of OLS results for short birth interval in Ethiopia, EDHS 2016.**

| Variable | Coefficient | Standard error | t-statistics | Probability | Robust std-error | Robust t-statistic | Robust probability | VIF |
|---|---|---|---|---|---|---|---|---|
| Intercept | 0.01 | 0.01 | 1.55 | 0.121 | 0.01 | 2.95 | <0.01 | ---------- |
| Women with no education | 0.50 | 0.02 | 24.87 | <0.01 | 0.01 | 31.98 | <0.01 | 1.34 |
| Women with primary education | 0.47 | 0.04 | 11.42 | <0.01 | 0.05 | 9.11 | <0.01 | 1.21 |
| Husbands with a higher education | 0.33 | 0.03 | 9.98 | <0.01 | 0.04 | 7.43 | <0.01 | 1.07 |
| Poorer wealth index | 0.30 | 0.05 | 5.61 | <0.01 | 0.07 | 4.18 | <0.01 | 1.46 |
| Middle wealth index | -0.48 | 0.05 | -9.41 | <0.01 | 0.06 | -7.88 | <0.01 | 1.43 |
| **OLS Diagnostics** | | | | | | | | |
| Number of observation: | 620 | | Akaike's Information Criterion (AICc): | | -445.28 | | | |
| Multiple R-squared: | 0.57 | | Adjusted R-Squared: | | 0.57 | | | |
| Joint F-Statistics: | 163.45 | | Prob(> F), (5,616) degrees: | | <0.01 | | | |
| Joint Wald Statistic: | 1646.99 | | Prob(> chi-squared), (5) degrees of freedom: | | <0.01 | | | |
| Koenker (BP) Statistics: | 82.22 | | Prob(> chi-squared), (5) degrees of freedom: | | <0.01 | | | |
| Jarque-Bera Statistics: | 0.63 | | Prob(> chi-squared), (2) degrees of freedom: | | 0.73 | | | |

*VIF* = Variance Inflation Factor; *EDHS* = Ethiopia Demographic and Health Survey.

education, having a husband with higher education, and coming from a poorer or middle wealth quintile.

## Geographic weighted regression (GWR)

The OLS regression identified predictors of short birth interval hot spots. However, it is a global model that assumes the relationship between each explanatory variable and short birth interval is stationary across the study area [55]. Table 3 depicts the GWR model for short birth interval in the study area. GWR improves the model fit when the relationship between the predictors and short birth interval is non-stationary [55, 57]. For instance, the adjusted $R^2$ value obtained from OLS increased from 0.57 (Table 2) to 0.64 using GWR (Table 3). In contrast to a higher corrected Akaike's Information Criterion value (AICc, which was -445.28) obtained from the OLS model (Table 2), the model from GWR provided a smaller AICc (-537.00; Table 3). Comparing the GWR AICc value to the OLS AICc value is one way to assess the

**Table 3. Geographic weighted regression (GWR) model for the short birth interval in Ethiopia, EDHS 2016.**

| Explanatory variables | Women with no education, women with primary education, husbands with higher education, poorer, and middle wealth quintiles |
|---|---|
| Residual squares | 13.32 |
| Effective number | 54.68 |
| Sigma | 0.15 |
| Akaike's Information Criterion (AICc) | -537.00 |
| Multiple R-Squared | 0.67 |
| Adjusted R-Squared | 0.64 |

*EDHS* = Ethiopia Demographic and Health Survey.

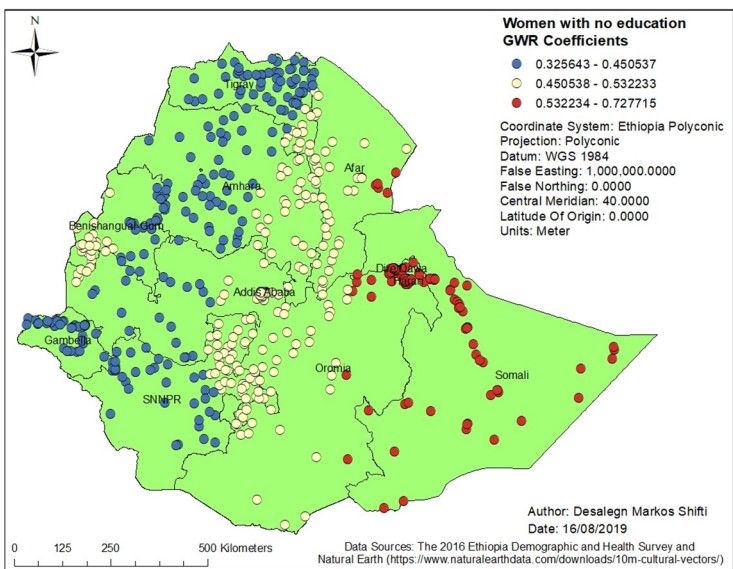

**Fig 3. Women with no education GWR coefficients for predicting short birth interval in Ethiopia, EDHS 2016.**
*GWR* = Geographically Weighted Regression; *SNNPR* = Southern Nation, Nationalities and People's region, *EDHS* = Ethiopia Demographic and Health Survey.

benefits of moving from a global model (OLS) to a local regression model (GWR). If the AICc values for two models (OLS and GWR) differ by more than 3, the model with the lower AICc is held to be better [58]. Overall, this study revealed that GWR analysis improved the model compared to the model estimated using OLS.

Figs 3, 4, 5, 6 and 7 demonstrate the geographic areas where the explanatory variables were strong and weak predictors of short birth interval. Not attending formal education by women,

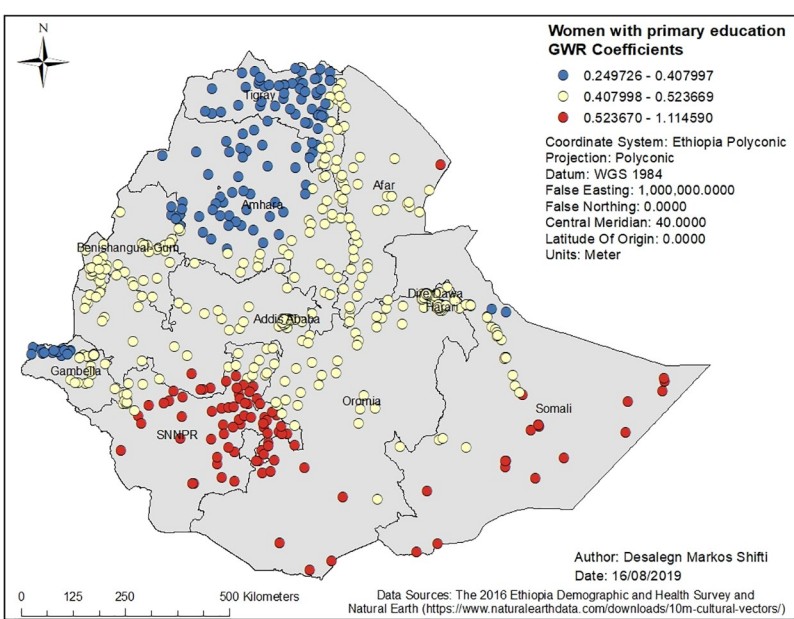

**Fig 4. Women with primary education GWR coefficients for predicting short birth interval in Ethiopia, EDHS 2016.** *GWR* = Geographically weighted Regression; *SNNPR* = Southern Nation, Nationalities and People's region, *EDHS* = Ethiopia Demographic and Health Survey.

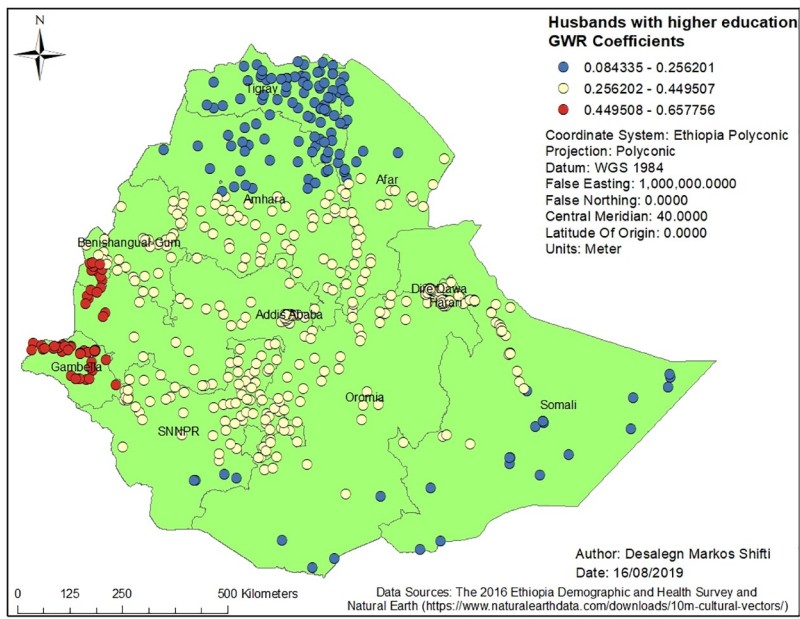

**Fig 5. Husbands with a higher education level of GWR coefficients for predicting short birth interval in Ethiopia, EDHS 2016.** *GWR* = Geographically Weighted Regression*; SNNPR* = Southern Nation, Nationalities and People's region, *EDHS* = Ethiopia Demographic and Health Survey.

for instance, had a positive relationship with short birth interval. As the proportion of women who had not attended formal education increased, the occurrence of short birth interval in Somali Region and Dire Dawa increased. In Fig 3, the geographic area with red-colored clustered points indicates where the coefficient for the variable not attending formal education is

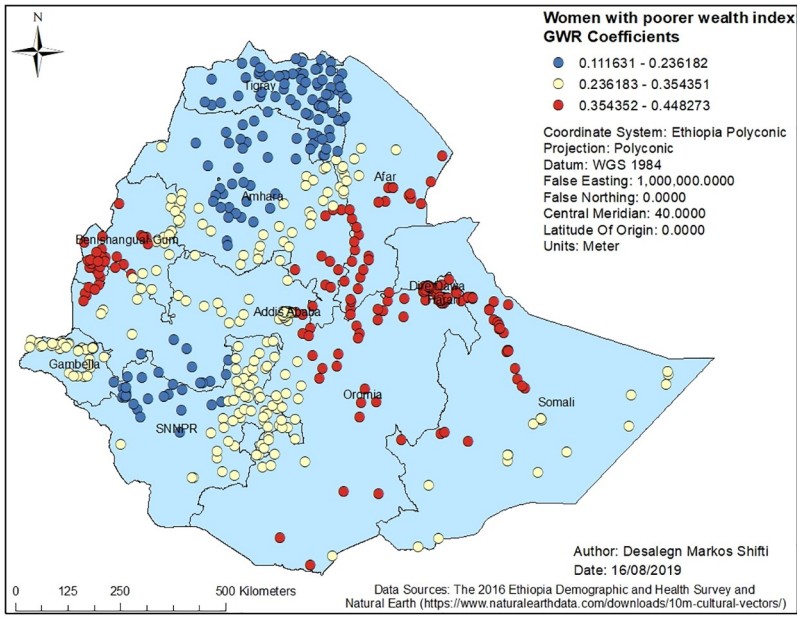

**Fig 6. Poorer wealth index GWR coefficients for predicting short birth interval in Ethiopia, EDHS 2016.** *GWR* = Geographically Weighted Regression*; SNNPR* = Southern Nation, Nationalities and People's region, *EDHS* = Ethiopia Demographic and Health Survey.

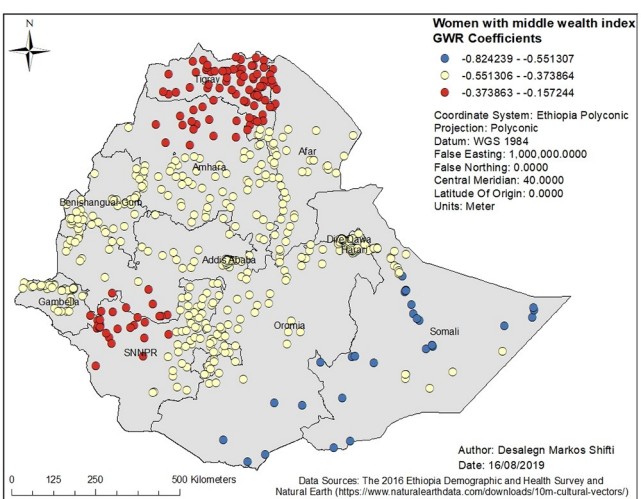

**Fig 7. Middle wealth index GWR coefficients for predicting short birth interval in Ethiopia, EDHS 2016.** *GWR* = Geographically Weighted Regression*; SNNPR* = Southern Nation, Nationalities and People's region, *EDHS* = Ethiopia Demographic and Health Survey.

largest. The larger the coefficient, the stronger the relationship [49]. Similarly, women with primary education was a strong predictor of short birth interval in the SNNPR and Somali Region. Fig 4 shows the coefficient of women who had attended primary education. Women having husbands who had attended higher education showed a strong and positive relationship with short birth interval in Gambella region and western Oromia Region. On the other hand, a weaker and positive relationship between husbands' higher level of education and short birth interval was observed in the Tigray Region and parts of Amhara Region. Fig 5 shows the coefficients for husbands who had attended a higher-level education. Fig 6 demonstrates the coefficients for poorer wealth quintile, which was found to be a key explanatory variable for predicting spatial variation in short birth interval. This variable has a strong positive correlation with short birth interval in Oromia Region, Benishangul-Gumuz Region, Dire Dawa, Afar Region and most parts of Somali Region. In contrast, clusters with a larger proportion of households from the middle wealth quintile were associated with a reduced occurrence of short birth interval. Fig 7 presents a map of the coefficients for the middle wealth quintile, indicating that being from the middle wealth quintile household was a negative and strong predictor of short birth interval in Tigray Region, parts of Amhara Region, and a few parts of SNNPR. It suggests that as the proportion of households in the middle wealth quintile in particular geographic areas increased, the occurrence of short birth interval in that area decreased. Similarly, middle wealth quintile was a moderate and negative predictor of short birth interval in the majority of Amhara Region, Oromia Region, Benishangul-Gumuz Region, Gambella, Addis Ababa, Dire Dawa, Afar, and a few parts of SNNPR, whereas it was a weak and negative predictor of short birth interval in most parts of Somali Region.

## Discussion

This is the first study to demonstrate the geographic variation of short birth interval and its associated predictors in Ethiopia. Short birth interval hot spots were observed in eastern Ethiopia (Somali Region and limited parts of Afar Region), northern Ethiopia (some parts of Oromia Region), and northwestern Ethiopia (some parts of SNNPR). It was found that women who had no education or primary education, husbands with higher education, and being from

a poorer or middle wealth quintile were predictors of short birth interval hot spots. The GWR coefficients of the two strong predictors of short birth interval across Ethiopia, for instance, range from 0.325643 to 0.727715 for women with no education and from 0.24926 to 1.114590 for women with primary education. The findings of the current study can be used to prioritize the expansion of family planning services to the most affected areas of Ethiopia.

The presence of statistically significant spatial variation in short birth interval across Ethiopian regions in our study is consistent with the finding of a study conducted in the DRC [14], which revealed the presence of statistically significant spatial variation of short birth interval across provinces. The observed geographic variation of short birth interval could be associated with a low level of modern contraceptive utilization in Afar Region and Somali Region, in addition to the predictors identified in the current study. This idea is supported by the findings of a previous study for cold spots (low proportion) of modern contraceptive utilization in these regions [59]. Additionally, the variation in short birth interval hot spots could be due to differences in demographic, cultural, and socioeconomic factors among regions. The majority of populations in Somali Region and Afar Region, for instance, are pastoralists, a lifestyle characterized by seasonal mobility [59]. People in these areas have limited access to health information and services. In addition to their mobility, pastoralist communities also live in very traditional settings and adhere strongly to cultural and religious values [60]. For instance, literature has documented that because of a fear of disapproval by community members due to cultural and religious norms, women in pastoralist areas were less likely to use modern contraceptives, which in turn affects birth interval duration [61, 62].

Not attending formal education by women had a strong positive relationship with hot spots of short birth interval in Somali Region and Dire Dawa. Similarly, women attending primary education only had a strong positive relationship with hot spots of short birth interval in SNNPR and Somali Region. Literature has also reported that the highest proportion of women with no education is in Somali Region (75.0%) and the lowest proportion is in Addis Ababa (9.0%) [15]. Education may influence birth interval by influencing women's health-seeking behavior, including the use of family planning services. In addition, women with higher education are more likely to have a higher level of health awareness, greater knowledge of available health services, improved ability to afford the cost of medical health care, and greater autonomy in making health-related decisions, including choices in family planning [63, 64].

The current study identified a positive relationship between husbands having a higher level of education and cold spots of short birth interval in Tigray Region and Amhara Region. It is possible that husbands with higher education possess knowledge of family planning methods, might encourage their partner to seek family planning services, and support optimum birth interval practice, which might account for the low proportion of short birth interval (cold spots) in Tigray Region and Amhara Region. On the other hand, women whose husbands had attended a higher level of education had a positive relationship with the hot spots of short birth interval in Gambella Region and western Oromia Region. The possible reasons for the observed finding in Gambella Region could be due to the low contraceptive prevalence rate in the region [59]. Further in-depth qualitative study is required to understand the finding in the western Oromia Region.

There was a positive and strong correlation between poorer wealth index of the household in the cluster and hot spot of short birth interval in most of Oromia Region, Afar Region, and Somali Region. Lack of adequate wealth in a community is likely to be associated with poor access to health information, and low utilization of maternal health care. The existing literature shows clear connections between household- and community-level socioeconomic factors and maternal health care utilization [65]. In addition, evidence has reported that wealth-based inequalities in contraceptive, which is a method to control birth interval use, still prevail in

Ethiopia despite efforts made to provide access to free contraception at all public health facilities [66].

It was found that as the proportion of households in the middle wealth quintile in geographic areas increased, the occurrence of short birth interval decreased. A household with a middle wealth quintile was a strong and negative predictor of short birth interval in Tigray Region, a few parts of Amhara Region and SNNPR. It is possible that women from middle wealth quintile households could relatively have better financial freedom to access and use family planning services and thus have optimum birth intervals. Evidence has also shown that disparities in access to family planning, knowledge of the services and direct contact with field workers are associated with the wealth gradient [67].

In this study, it was found that different predictors of short birth interval (women with no education or with primary education, having a husband with higher education, and coming from a household with a poorer wealth quintile or middle wealth quintile) act more/less strongly across regions. For instance, women not attending formal education or attending primary education only had a strong positive relationship with hot spots of short birth interval in Somali Region while they had a week positive association with short birth interval in Amhara and Tigray region. This might be related to variations in the availability and accessibility of different family planning services, socioeconomic and cultural variations across administrative regions of the country.

The findings of this study were not without limitations. The geographical coordinates of clusters were displaced by up to 2km in urban areas, 5km for most clusters in rural areas and 10km for 1% of clusters in rural areas to prevent identification of respondents or the community; this could affect estimated cluster effects in the spatial analysis. In addition, the EDHS did not collect data regarding the maternal previous history (before the index child) of multiple pregnancies, caesarean section, and comorbidities, which could be potential predictors for the spatial variation of short birth interval. Therefore, the interpretation or conclusion based on this study should consider these limitations. Despite the above limitation, the study has the following strengths. First, it used data from the most recent, nationally representative and largest sample of a population-based survey, which covers all regions and administrative cities of the country. In addition, it used a robust methodology to analyze the data closely to assess the non-stationary nature of predictors.

## Conclusion

Overall, the study illustrated that there was a spatial variation of short birth interval among women in Ethiopia. Short birth interval hot spots were identified in Somali Region, Oromia Region, SNNPR and a few parts of Afar Region. Being a resident in a geographic area with a high proportion of women having no education or primary education, husbands who had attended higher education and households from a poorer or middle wealth quintile increased the risk of experiencing short birth interval.

National and regional decision makers should give priority to the identified hot spot clusters through further expansion of family planning services. The government should involve religious and community leaders in efforts to reduce the magnitude of short birth interval. Working with religious leaders has been shown to be one of the emerging effective strategies, and is associated with increasing modern family planning use [68, 69]. Likewise, community interventions targeting empowering women with education help reduce the magnitude of short birth interval. Inter-sectoral collaboration between the Ministry of Health and Ministry of Education is required to improve women's literacy status. In addition, using mass media to disseminate consistent messages about the WHO recommended birth interval practice could

help increase awareness of the community regarding the benefits of an optimal birth interval. Moreover, initiating community dialogue to create awareness about optimum birth interval is recommended. Integrating birth interval counseling, as per the WHO recommendation, with antenatal care, postnatal care, family planning, and immunization services should be supported. Further studies that assess cultural and social factors affecting the geographic variation of short birth interval is also vital.

## Implications for program planners and policy practice

Our detailed maps of short birth interval hot spots and its predictors enable decision makers to exercise precision public health. For instance, the findings of this study can help policy makers and health programmers set national and regional targets to reduce the magnitude of short birth interval in a defined period. In addition, the current study identified target areas where resources should be directed to prevent short birth interval. Moreover, the evidence obtained from the current study can be used to monitor the impact of family planning programs across regions of the country. Another inference that can guide policy makers and program planners which emerged from this study is that the association between short birth interval hot spots and significant explanatory variables (women having no education or primary education, having husbands with a higher education, and being from a poorer or middle wealth quintile) varied across administrative regions of the country. Consequently, health policy and intervention directed towards reducing the burden of short birth interval should focus on the local association of the abovementioned predictors of short birth interval hot spots. Targeted interventions such as empowering women through education and improving household income are among the key strategies to promote optimum birth interval practice of the population.

## Supporting information

**S1 Table. Candidate explanatory variables used in the Exploratory Regression tool.** (DOCX)

## Acknowledgments

We are grateful to The DHS Program for allowing us to use the Ethiopia Demographic and Health Survey (EDHS) data for further analysis.

## Author Contributions

**Conceptualization:** Desalegn Markos Shifti, Catherine Chojenta, Elizabeth G. Holliday, Deborah Loxton.

**Data curation:** Desalegn Markos Shifti.

**Formal analysis:** Desalegn Markos Shifti.

**Investigation:** Desalegn Markos Shifti.

**Methodology:** Desalegn Markos Shifti, Catherine Chojenta, Elizabeth G. Holliday, Deborah Loxton.

**Resources:** Desalegn Markos Shifti.

**Software:** Desalegn Markos Shifti.

**Supervision:** Catherine Chojenta, Elizabeth G. Holliday, Deborah Loxton.

**Writing – original draft:** Desalegn Markos Shifti.

**Writing – review & editing:** Desalegn Markos Shifti, Catherine Chojenta, Elizabeth G. Holliday, Deborah Loxton.

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
