## [Decision Letter · Decision Letter 0]

12 Feb 2020

PONE-D-19-32257

Application of Geographically Weighted Regression analysis to assess predictors of short birth interval hot spots in Ethiopia

PLOS ONE

Dear Mr. Shifti,

Thank you for submitting your manuscript to PLOS ONE. After careful consideration, we feel that it has merit but does not fully meet PLOS ONE’s publication criteria as it currently stands. Therefore, we invite you to submit a revised version of the manuscript that addresses the points raised during the review process.

Please see comments from both reviewers below. Looking forward to reviewing your revised manuscript.

We would appreciate receiving your revised manuscript by Mar 28 2020 11:59PM. To enhance the reproducibility of your results, we recommend that if applicable you deposit your laboratory protocols in protocols.io, where a protocol can be assigned its own identifier (DOI) such that it can be cited independently in the future. For instructions see: http://journals.plos.org/plosone/s/submission-guidelines#loc-laboratory-protocols

We look forward to receiving your revised manuscript.

Kind regards,

Charles A. Ameh, PhD, MPH, FWACS (OBGYN), FRCOG

Academic Editor

PLOS ONE

Journal Requirements:

1. We note that [Figure(s) 2-7] in your submission contain [map/satellite] images which may be copyrighted. All PLOS content is published under the Creative Commons Attribution License (CC BY 4.0), which means that the manuscript, images, and Supporting Information files will be freely available online, and any third party is permitted to access, download, copy, distribute, and use these materials in any way, even commercially, with proper attribution. For these reasons, we cannot publish previously copyrighted maps or satellite images created using proprietary data, such as Google software (Google Maps, Street View, and Earth). For more information, see our copyright guidelines: http://journals.plos.org/plosone/s/licenses-and-copyright.

1.    You may seek permission from the original copyright holder of Figure(s) [2-7] to publish the content specifically under the CC BY 4.0 license. 

Additional Editor Comments (if provided):

Thanks for a very interesting manuscript, please see comments from both reviewers. Looking forward to reviewing your revised manuscript.

Reviewers' comments:

Reviewer's Responses to Questions

**Comments to the Author**

1. Is the manuscript technically sound, and do the data support the conclusions?

Reviewer #1: Yes

Reviewer #2: Partly

2. Has the statistical analysis been performed appropriately and rigorously? 

Reviewer #1: Yes

Reviewer #2: Yes

3. Have the authors made all data underlying the findings in their manuscript fully available?

Reviewer #1: Yes

Reviewer #2: Yes

4. Is the manuscript presented in an intelligible fashion and written in standard English?

Reviewer #1: Yes

Reviewer #2: Yes

5. Review Comments to the Author

Reviewer #1: The authors have done a commendable job analysis the spatial variability of predictors’ effects on short birth interval across Ethiopia.

Abstract

Overall well-written abstract, however, I suggest including the meaning of “short” in “short birth interval” in the background/methods section

Methods

• Did any women have more than >2 births during the recall period? How were these dealt with in the analysis?

• Add description of how GPS data is collected in the DHS.

• Line 48: the definition of what varies?

• Line 103: because birth interval is the focus of this analysis, please give a short explanation of how birth interval data were collected in the manuscript, on top of directing readers to the DHS Interviewer’s Manual for further details.

• Line 118: I think the authors might be referring to sampling weights here, either way, please specify what weights were applied.

• Lines 168-169: Positive, instead of negative, statements should be easier to understand. Rephrase to something along the line of “The model should unbiased…. The residuals should normally distributed and present no spatial patterns.”

Results

• Line 206: This statement may be misinterpreted, as majority of all births in Ethiopia are in rural areas. Either make this point clear, or present % of short birth intervals by place of residence.

• What is the Getis-Ord General G statistic found? What is the distance threshold used for hot-spot analysis?

• Lines 266-270: Why did the authors select the middle wealth quintile for comparison? The proportion of households in the middle wealth quintile of an EA can be higher (leading to a decreased occurrence of short birth interval, as shown here) either for EAs that are relatively poorer/wealthier. This makes the effect of socioeconomic status less apparent.

Discussion

• First paragraph: I suggest adding the important finding of the variable strengths of the different predictors on short birth interval across Ethiopia.

• Second paragraph: Data on both religion and ethnicity can collected in the DHS. Why have they not been considered in the current analysis?

• Lines 309-317: And what is the authors’ interpretation of between higher husband education and hot spots of birth interval in Oromia Region, SNNPR and Somali Region?

• The discussion will benefit from the authors’ interpretation of the mechanisms through which different predictors act more/less strongly in different places in the country.

• In Ethiopia, increased uptake of contraceptives has not translated into reduced birth interval, which may indicate additional unmet need (for limiting and for spacing) among women who have given birth, perhaps due to lack of postpartum counselling or other reasons. The authors may want to comment on this in the current manuscript.

Reviewer #2: This is a very interesting and, overall, potentially useful piece of work from the health services planning perspective. I would recommend its publication should the authors clarify some questions. Briefly stated:

Major considerations

Despite the moderate coefficient of determination and the values of the statistics used to assess the goodness-of-fit of the different models, I wonder why the authors did not considered (factored in) the following covariates in the models given their likelihood of being associated with (explanatory o predictive variables) short-birth interval: previous 1) multiple pregnancies, 2) miscarriages, 3) c-sections, and 4) previous and current comorbidities. If some of them were associated, the models would incur in an important omitted variable bias, and reported coefficients and results could be biased. The authors must first account for this, and depending on the justification they must then explain in the Discussion the omission and its likely effects on the results.

Minor considerations

Lines 97-99. Women who had never been married were not included in the study, since women who have multiple births out of wedlock are unlikely to plan their births in the same way as married women.

What is the share of women who have never been married in Ethiopia and the different regions? How do women with multiple births out of wedlock plan their births? A substantial share could be biasing the results. This being the casa it should be addressed in the Discussion. Otherwise, it should be so stated.

Lines 79-80. Strictly speaking, the formulation of the objective of the study (The current study aimed to assess predictors of short birth interval hot spots in Ethiopia) should not include the methods used to accomplish it (by using Geographically Weighted Regression (GWR) analysis.).

6. PLOS authors have the option to publish the peer review history of their article (what does this mean?). If published, this will include your full peer review and any attached files.

Reviewer #1: Yes: Kerry LM Wong

Reviewer #2: No

---

## [Author Response · Author response to Decision Letter 0]

10 Mar 2020

Editorial comments: Please ensure that your manuscript meets PLOS ONE's style requirements, including those for file naming. The PLOS ONE style templates can be found at http://www.journals.plos.org/plosone/s/file?id=wjVg/PLOSOne_formatting_sample_main_body.pdf and http://www.journals.plos.org/plosone/s/file?id=ba62/PLOSOne_formatting_sample_title_authors_affiliations.pdf

Response: Our manuscript has been revised to meet PLOS ONE's style requirements.

Editorial comments: 1. We note that [Figure(s) 2-7] in your submission contain [map/satellite] images which may be copyrighted. All PLOS content is published under the Creative Commons Attribution License (CC BY 4.0), which means that the manuscript, images, and Supporting Information files will be freely available online, and any third party is permitted to access, download, copy, distribute, and use these materials in any way, even commercially, with proper attribution. For these reasons, we cannot publish previously copyrighted maps or satellite images created using proprietary data, such as Google software (Google Maps, Street View, and Earth). For more information, see our copyright guidelines: http://journals.plos.org/plosone/s/licenses-and-copyright.

1. You may seek permission from the original copyright holder of Figure(s) [2-7] to publish the content specifically under the CC BY 4.0 license. 

Natural Earth (public domain): http://www.naturalearthdata.com/ Response: The map of Ethiopia (i.e., Figure 2-7) that was used to develop hot and/or cold spots of short birth interval has been used under the terms of use of the Natural Earth (public domain) (https://www.naturalearthdata.com/about/terms-of-use/), which permits to use the maps in any manner, including modifying the content and design, electronic dissemination, and offset printing. In addition, the geographic coordinates (latitude and longitude) of the clusters in the map were used after getting permission from The DHS Program (https://dhsprogram.com/data/available-datasets.cfm) to use the Ethiopia Demographic and Health Survey data for further analysis and all data are available online upon request. 

The above-mentioned two sources have been acknowledged in all figures (Figure 2-7). Furthermore, the source of the map has been mentioned in the method section of this revised version of the manuscript as follows (line 109-112). 

‘In addition, the administrative polygons of Ethiopia, which were obtained from the Natural Earth [35] has been used to develop the map of hot and/or cold spots of short birth interval. The country’s administrative polygons reflect administrative boundaries, such as regions, zones, and districts of Ethiopia.’

Reviewer #1: The authors have done a commendable job analysis the spatial variability of predictors’ effects on short birth interval across Ethiopia.

Response: Thank you.

Reviewer #1: Abstract

Overall well-written abstract, however, I suggest including the meaning of “short” in “short birth interval” in the background/methods section

Response: We have now incorporated the meaning of ‘short’ in “short birth interval” in the background section of the abstract (line 18-19).

Reviewer #1: Did any women have more than >2 births during the recall period? How were these dealt with in the analysis?

Response: When women had more than two births within the five years preceding the survey, birth interval of the most recent births were considered in this current study. The 2016 Ethiopia Demographic and Health Survey report that has been released to the public also followed this approach. 

In this revised version of the manuscript, we have included a statement describing the inclusion of the most recent birth in the method section of this revised version of the manuscript as follow (line 98-100): 

‘When women had more than two births in the five years preceding the survey, birth interval of the most recent two births was considered in this current study.’

Reviewer #1: Add description of how GPS data is collected in the DHS.

Response: We have incorporated information regarding GPS data collection accordingly (line 101-109):

‘Global Positioning System (GPS) receivers were used to collect the location data (geographic coordinates) of each survey cluster. The GPS reading was made at the centre of each cluster. The GPS data collectors ensured the centre was relatively open, away from tall buildings, and out from under tree canopy in order to receive adequate satellite signal strength. To maintain respondents’ confidentiality, GPS latitude/longitude positions for all survey clusters were randomly displaced. The maximum displacement for urban clusters was two kilometres (km) and five km for 99% of rural clusters. The remaining 1% of the rural clusters were displaced a maximum of 10 km. The displacement was restricted to the country's second administrative level (DHS survey region) so that the points stay within the country [34].’

Reviewer #1: Line 48: the definition of what varies?

Response: The sentence has been rewritten as follow (line 49):

‘A short birth interval is associated with a higher risk of maternal morbidities, including miscarriage, preeclampsia, high blood pressure, and premature rupture of membranes (4, 5).’

Reviewer #1: Line 103: because birth interval is the focus of this analysis, please give a short explanation of how birth interval data were collected in the manuscript, on top of directing readers to the DHS Interviewer’s Manual for further details.

Response: We have incorporated information regarding how birth interval data were collected (line 115-125).

‘Women’s birth interval data were collected through reviewing the date of birth of their biological children from children’s birth /immunization certificate and/or asking information regarding their children’s date of birth from the women. Birth interval data of women for all their children born live irrespective of their survival status at the time of the interview were collected. For children who had birth certificates, their mothers were asked to confirm the accuracy of the information prior to documenting children’s date of birth. This was done to avoid errors because in some cases the information on the document may be the date when the birth was recorded and not the date when the child was born. When children did not have a birth certificate, information regarding their date of birth were obtained from their mothers. Then, the length of birth interval was computed in months and the data were accessible for further analysis in this form.’

Reviewer #1: Line 118: I think the authors might be referring to sampling weights here, either way, please specify what weights were applied.

Response: We have now specified the type of weight used as follows (line 141).

‘Before performing spatial analysis, the weighted proportion (using sample weight) of short birth interval and candidate explanatory variables (see supplementary Table 1) data were exported to ArcGIS.’

Reviewer #1: Lines 168-169: Positive, instead of negative, statements should be easier to understand. Rephrase to something along the line of “The model should unbiased…. The residuals should normally distributed and present no spatial patterns.”

Response: The comment is accepted and we have corrected the statement accordingly (line 192).

‘The model should be unbiased (heteroscedasticity or non-stationarity). The residuals should be normally distributed and revealed no spatial patterns.’

Reviewer #1: Line 206: This statement may be misinterpreted, as majority of all births in Ethiopia are in rural areas. Either make this point clear, or present % of short birth intervals by place of residence.

Response: We are afraid that the statement was not misinterpreted (line 229). We had already computed the column percentage and presented the percentage of short birth interval by place of residence in the previous version of the manuscript. The finding of the study showed that the majority (94.0%) of women with short birth interval in Ethiopia resided in rural areas. Would you please see the statement again? 

‘Among women who experienced short birth interval, 94.0% resided in a rural part of the country.’

For further information, please refer to Table 1.

Reviewer #1: What is the Getis-Ord General G statistic found? What is the distance threshold used for hot-spot analysis?

Response: We have included the finding of Getis-Ord General G test and the distance threshold used for hot-spot analysis as follow (line 236-239): 

‘The Getis-Ord General G statistic revealed the presence of high clustering (z-score=4.328, p-value<0.001). The average distance at which a feature (cluster in this case) has at least one neighbour was 18 km. The maximum distance at which clustering of short birth interval rate peaked was at 122 km.’

Reviewer #1: Lines 266-270: Why did the authors select the middle wealth quintile for comparison? The proportion of households in the middle wealth quintile of an EA can be higher (leading to a decreased occurrence of short birth interval, as shown here) either for EAs that are relatively poorer/wealthier. This makes the effect of socioeconomic status less apparent.

Response: At the variable selection phase of our analysis (using Exploratory Regression tool), it was only the middle wealth quintile that fulfilled the assumption of the ordinary least squares (OLS) (line 188-192). That is why it was included in the model. As it is described in the method section (line 127-132), each wealth quintile was among the candidate explanatory variables that were considered in the Exploratory Regression of the current study (also see S1 Table).

Reviewer #1: First paragraph: I suggest adding the important finding of the variable strengths of the different predictors on short birth interval across Ethiopia. 

Second paragraph: Data on both religion and ethnicity can collected in the DHS. Why have they not been considered in the current analysis?

Response: We accept the comment and have incorporated those findings accordingly (line 305-308): 

‘The GWR coefficients of the two strong predictors of short birth interval across Ethiopia, for instance, range from 0.325643 to 0.727715 for women with no education and from 0.24926 to 1.114590 for women with primary education.’ 

Yes, data on both religion and ethnicity were collected in the DHS. However, in our study, the variables did not satisfy the criteria of properly specified Ordinary Least Squares (OLS) models in the Exploratory Regression tool (line 188-194). On the other hand, those predictors identified in the current study [i.e., women’s education, husband’s education, and wealth quintile] were the ones with the highest adjusted R2 (adjusted R2=0.64) and met all the requirement of OLS. A search of peer-reviewed articles and grey literature showed a dearth of information on the association between ethnicity and birth interval experience in Ethiopia. The authors have also recommended (line 404-405) for further studies that assess the cultural factors affecting the geographic variation of short birth interval.

Reviewer #1: Lines 309-317: And what is the authors’ interpretation of between higher husband education and hot spots of birth interval in Oromia Region, SNNPR and Somali Region?

Response: 

We have refined our previous justification and incorporated the additional reasons for the observed finding as follows (line 337-346). 

‘It is possible that husbands with higher education possess knowledge of family planning methods, might encourage their partner to seek family planning services, and support optimum birth interval practice, which might account for the low proportion of short birth interval (cold spots) in Tigray Region and Amhara Region. On the other hand, women whose husbands had attended a higher level of education had a positive relationship with the hot spots of short birth interval in Gambella Region and western Oromia Region. The possible reasons for the observed finding in Gambella Region could be due to the low contraceptive prevalence rate in the region [58]. Further in-depth qualitative study is required to understand the finding in the western Oromia Region.’

Reviewer #1: The discussion will benefit from the authors’ interpretation of the mechanisms through which different predictors act more/less strongly in different places in the country.

Response: We agree with the reviewer. In addition to the implication of each finding of the current study discussion for each predictor of short birth interval, we have added the following statement (line 364-372): 

‘In this study, it was found that different predictors of short birth interval (women with no education or with primary education, having a husband with higher education, and coming from a household with a poorer wealth quintile or middle wealth quintile) act more/less strongly across regions. For instance, women not attending formal education or attending primary education only had a strong positive relationship with hot spots of short birth interval in Somali Region while they had a week positive association with short birth interval in Amhara and Tigray region. This might be related to variations in the availability and accessibility of different family planning services and cultural variation across administrative regions of the country.’

Reviewer #1: In Ethiopia, increased uptake of contraceptives has not translated into reduced birth interval, which may indicate additional unmet need (for limiting and for spacing) among women who have given birth, perhaps due to lack of postpartum counselling or other reasons. The authors may want to comment on this in the current manuscript.

Response: 

The information that shows the increase in uptake of contraceptives in Ethiopia over time but the unchanged birth interval has been obtained from previous literature (line 56-59; under the introduction section). 

‘Moreover, despite the increasing trend in contraceptive utilization in Ethiopia, from 5.9% in 2000 to 14.0% in 2005, 27.0% in 2011 and 35.0% in 2016, the prevalence of short birth interval has remained unchanged, 19.7% in 2000, 21.4% in 2005, 20.5% in 2011 and 21.7% in 2016 [15-18].’

There could be individual-, household- and community-level factors that are responsible for the above-mentioned findings. Since the objective of the current study was to assess the predictors of short birth interval hot spots in Ethiopia, and it is beyond the scope of the current study to comment on that. 

Literature has also shown that modern contraceptives are effective at preventing conception; what is not understood is why, in practice, their use may not always result in longer birth intervals or preventing short birth interval (Yeakey MP, et al., 2009. How Contraceptive Use Affects Birth Intervals: Results of a Literature Review. Family Planning; 40(3): 205–214). Even though women who use contraceptives increase over time unless they took the contraception to achieve an optimum birth interval (as per the WHO recommendation) the magnitude of short birth interval may not be reduced. It could also be associated with improper contraceptive use, which may end up with unintended/unplanned pregnancy and as a result a short birth interval. 

Generally, the issue of why the increase in trend of contraception use over time in Ethiopia did not result in reducing the magnitude of short birth interval (described in the introduction section; line 56-59) is complex and beyond the scope of the current study. The authors believe that it requires further studies (it could be trend analysis and assessing its associated factors) to comment on the underlying factors responsible for the observed findings.

We also would like to thank you for raising this issue and will consider it for further study in the future. 

Reviewer #2: Is the manuscript technically sound, and do the data support the conclusions?

Response of reviewer #2: Partly

Response: For further clarification, the conclusion has been rewritten as follow (line 386-387): 

‘Overall, the study illustrated that there was a spatial variation of short birth interval among women in Ethiopia. Short birth interval hot spots were identified in Somali Region, Oromia Region, SNNPR and a few parts of Afar Region. Being a resident in a geographic area with a high proportion of women having no education or primary education, husbands who had attended higher education and households from a poorer or middle wealth quintile increased the risk of experiencing short birth interval.’

Reviewer #2: This is a very interesting and, overall, potentially useful piece of work from the health services planning perspective. I would recommend its publication should the authors clarify some questions. Briefly stated

Response: Thank you.

Reviewer #2: Major considerations

Despite the moderate coefficient of determination and the values of the statistics used to assess the goodness-of-fit of the different models, I wonder why the authors did not considered (factored in) the following covariates in the models given their likelihood of being associated with (explanatory o predictive variables) short-birth interval: previous 1) multiple pregnancies, 2) miscarriages, 3) c-sections, and 4) previous and current comorbidities. If some of them were associated, the models would incur in an important omitted variable bias, and reported coefficients and results could be biased. The authors must first account for this, and depending on the justification they must then explain in the Discussion the omission and its likely effects on the results.

Response: We appreciate your observation. In order to consider these particular variables (i.e., previous history of multiple pregnancies, miscarriages, cesarean section, and previous and current comorbidities) as potential factors, data regarding the presence/absence of the aforementioned events before the birth of the index child should be collected. However, these variables were not collected in the Ethiopia Demographic and Health Survey. In this revised version of the manuscript, the implication of not including some of these variables has been included. 

The authors, however, believe that it would not be relevant to consider the history of abortion as one of the potential predictors. Because the most recent WHO recommendation for a healthy pregnancy interval after abortion is six months (World Health Organization. Report of a WHO Technical Consultation on Birth Spacing. Geneva, Switzerland. 13-15 June 2005.), which could fall under short birth interval category while it is not in reality. Therefore, considering this variable as one of the potential factors could be misleading. 

Regarding comorbidities, it would have been good to include the comorbidity status of women at/after the birth of the preceding child (not at the birth of the index child) as a predictor. Nevertheless, the data were not available in the 2016 Ethiopia Demographic and Health Survey. As a result, it was not included in the analysis. 

To acknowledge some of the above limitations of our study, the following statement has been included in this revised version of the manuscript (line 376-380): 

‘In addition, the EDHS did not collect data regarding the maternal previous history (i.e., before the index child) of multiple pregnancies, caesarean section, and comorbidities, which could be potential predictors for the spatial variation of short birth interval. Therefore, the interpretation or conclusion based on this study should consider these limitations.’

Reviewer #2: Minor considerations

Lines 97-99. Women who had never been married were not included in the study, since women who have multiple births out of wedlock are unlikely to plan their births in the same way as married women.

What is the share of women who have never been married in Ethiopia and the different regions? How do women with multiple births out of wedlock plan their births? A substantial share could be biasing the results. This being the casa it should be addressed in the Discussion. Otherwise, it should be so stated.

Response: First of all, the total number of women who had never been married but eligible to provide birth interval data was small (n=12; 1.4%). With regard to their sample size, excluding these women is less likely to bias the findings of the study. 

Second, literature has shown that marital status of women had a contribution to the differences in the prevalence of postpartum family planning use among women (Gebremedhin AY. et al. Family planning use and its associated factors among women in the extended postpartum period in Addis Ababa, Ethiopia. Contraception and Reproductive Medicine (2018) 3:1). The study performed by Gebremedhin AY. et al., 2018 also pointed out that the difference in postpartum family planning use among married and not married women could be associated with their difference in risk perception related to unwanted or mistimed pregnancy, which is expected to be high among married women than none married ones. 

Finally, having multiple children out of wedlock in a culturally conservative society, like Ethiopia, is not common. Though it requires peer-reviewed evidence, the primary author knows the study area very well and understand the relationship between marital status and birth interval issue in the country. 

For further clarity, the number of women who had never been married and excluded from the analysis (n=12) was specified in the method section of this revised version of the manuscript (line 96).

Reviewer #2: Minor considerations

Lines 79-80. Strictly speaking, the formulation of the objective of the study (The current study aimed to assess predictors of short birth interval hot spots in Ethiopia) should not include the methods used to accomplish it (by using Geographically Weighted Regression (GWR) analysis.).

Response: We agree with the reviewer and have deleted the statement that specifies the methods used to accomplish the objective. It is rewritten as follows (line 79):

‘The current study aimed to assess predictors of short birth interval hot spots in Ethiopia.’

---

## [Editor Report · Decision Letter 1]

24 Apr 2020

PONE-D-19-32257R1

Application of Geographically Weighted Regression analysis to assess predictors of short birth interval hot spots in Ethiopia

PLOS ONE

Dear Mr. Shifti,

Thank you for submitting your manuscript to PLOS ONE. After careful consideration, we feel that it has merit but does not fully meet PLOS ONE’s publication criteria as it currently stands. Therefore, we invite you to submit a revised version of the manuscript that addresses the points raised during the review process.

We would appreciate receiving your revised manuscript by Jun 08 2020 11:59PM. To enhance the reproducibility of your results, we recommend that if applicable you deposit your laboratory protocols in protocols.io, where a protocol can be assigned its own identifier (DOI) such that it can be cited independently in the future. For instructions see: http://journals.plos.org/plosone/s/submission-guidelines#loc-laboratory-protocols

We look forward to receiving your revised manuscript.

Kind regards,

Charles A. Ameh, PhD, MPH, FWACS (OBGYN), FRCOG

Academic Editor

PLOS ONE

Additional Editor Comments (if provided):

Thanks for a comprehensive and satisfactory response to most of the reviewers comments. One questions is still unresolved: How did you deal with the analysis of women with more than 2 births during the recall period?

---

## [Author Response · Author response to Decision Letter 1]

11 May 2020

Thanks for the constructive feedback. When women had more than 2 births during the recall period (i.e., five years preceding the survey), birth interval of their most recent two births, which is birth interval between the index child and the immediately preceding child of women was uniformly considered for all the study participants. The 2016 Ethiopia DHS in its report released to the public also considered birth interval information from women’s most recent two births (i.e., the birth interval between the index child and the immediately preceding child) (Central Statistical Agency (CSA) [Ethiopia] and ICF. Ethiopia Demographic and Health Survey 2016. Addis Ababa, Ethiopia, and Rockville, Maryland, USA: CSA and ICF. 2016.). In this revised version of the manuscript (line 98-101), the inclusion of women’s birth interval from their most recent two births is described as follow: 

‘When women had more than two births in the five years preceding the survey, birth interval of their most recent two births (i.e., the birth interval between the index child and the immediately preceding child) was uniformly considered for all the study participants.’

All figures are uploaded to the Preflight Analysis and Conversion Engine (PACE) digital diagnostic tool, https://pacev2.apexcovantage.com/.

---

## [Editor Report · Decision Letter 2]

13 May 2020

Application of Geographically Weighted Regression analysis to assess predictors of short birth interval hot spots in Ethiopia

PONE-D-19-32257R2

Dear Dr. Shifti,

We are pleased to inform you that your manuscript has been judged scientifically suitable for publication and will be formally accepted for publication once it complies with all outstanding technical requirements.

With kind regards,

Charles A. Ameh, PhD, MPH, FWACS (OBGYN), FRCOG

Academic Editor

PLOS ONE

Additional Editor Comments (optional):

All comments have been satisfactorily addressed. Congratulations
---

## [Editor Report · Acceptance letter]

20 May 2020

PONE-D-19-32257R2 

Application of Geographically Weighted Regression analysis to assess predictors of short birth interval hot spots in Ethiopia 

Dear Dr. Shifti:

I am pleased to inform you that your manuscript has been deemed suitable for publication in PLOS ONE. Congratulations! Your manuscript is now with our production department. 

With kind regards,

on behalf of

Dr. Charles A. Ameh 

Academic Editor

PLOS ONE